# Fecal Metaproteomics Reveals Reduced Gut Inflammation and Changed Microbial Metabolism Following Lifestyle-Induced Weight Loss

**DOI:** 10.3390/biom11050726

**Published:** 2021-05-12

**Authors:** Ronald Biemann, Enrico Buß, Dirk Benndorf, Theresa Lehmann, Kay Schallert, Sebastian Püttker, Udo Reichl, Berend Isermann, Jochen G. Schneider, Gunter Saake, Robert Heyer

**Affiliations:** 1Institute of Laboratory Medicine, Clinical Chemistry and Molecular Diagnostics, Leipzig University, Paul-List-Str. 13/15, 04103 Leipzig, Germany; berend.isermann@medizin.uni-leipzig.de; 2Bioprocess Engineering, Otto von Guericke University, Universitätsplatz 2, 39106 Magdeburg, Germany; enricobuss14@yahoo.de (E.B.); theresa.schlegel@ovgu.de (T.L.); kay.schallert@ovgu.de (K.S.); Sebastian.puettker@gmx.de (S.P.); ureichl@mpi-magdeburg.mpg.de (U.R.); 3Microbiology, Anhalt University of Applied Sciences, Bernburger Straße 55, 06354 Köthen, Germany; 4Bioprocess Engineering, Max Planck Institute for Dynamics of Complex Technical Systems, Sandtorstraße 1, 39106 Magdeburg, Germany; 5Institute of Clinical Chemistry and Pathobiochemistry, Otto von Guericke University, Leipziger Straße 44, 39120 Magdeburg, Germany; 6Luxembourg Centre for Systems Biomedicine (LCSB), University of Luxembourg, 6, Avenue du Swing, L-4367 Belvaux, Luxembourg; jochen.schneider@uni.lu; 7Department of Internal Medicine II, Saarland University Medical Center, Kirrberger Str., 66424 Homburg Saar, Germany; 8Database and Software Engineering Group, Otto von Guericke University, Universitätsplatz 2, 39106 Magdeburg, Germany; saake@iti.cs.uni-magdeburg.de

**Keywords:** metaproteomics, fecal samples, obesity, metabolic syndrome, gut inflammation, microbiome, weight loss

## Abstract

Gut microbiota-mediated inflammation promotes obesity-associated low-grade inflammation, which represents a hallmark of metabolic syndrome. To investigate if lifestyle-induced weight loss (WL) may modulate the gut microbiome composition and its interaction with the host on a functional level, we analyzed the fecal metaproteome of 33 individuals with metabolic syndrome in a longitudinal study before and after lifestyle-induced WL in a well-defined cohort. The 6-month WL intervention resulted in reduced BMI (−13.7%), improved insulin sensitivity (HOMA-IR, −46.1%), and reduced levels of circulating hsCRP (−39.9%), indicating metabolic syndrome reversal. The metaprotein spectra revealed a decrease of human proteins associated with gut inflammation. Taxonomic analysis revealed only minor changes in the bacterial composition with an increase of the families Desulfovibrionaceae, Leptospiraceae, Syntrophomonadaceae, Thermotogaceae and Verrucomicrobiaceae. Yet we detected an increased abundance of microbial metaprotein spectra that suggest an enhanced hydrolysis of complex carbohydrates. Hence, lifestyle-induced WL was associated with reduced gut inflammation and functional changes of human and microbial enzymes for carbohydrate hydrolysis while the taxonomic composition of the gut microbiome remained almost stable. The metaproteomics workflow has proven to be a suitable method for monitoring inflammatory changes in the fecal metaproteome.

## 1. Introduction

The development of obesity and metabolic syndrome depends to a large extent on an individual’s unique metabolic processing of foods, genetics, lifestyle, and gut microbiome composition. Experimental and observational evidence suggests that obesity-associated inflammation plays a central role in metabolic dysfunction and disease progression. Recent studies indicate that gut microbiome-mediated inflammation may promote metabolic disorders and obesity-associated low-grade inflammation [1]. Accordingly, taxonomic and functional alterations of the gut microbiome are typically observed in inflammatory gut diseases [2] but also in metabolic diseases such as diabetes mellitus [3] or obesity [4,5]. In addition, fecal microbiota transplantation in mice even revealed that the microbiome might cause obesity [6].

Close spatial and functional interactions of the host and gut microbiota require a well-tuned balance. In particular, the host immune system must tolerate mutualistic bacteria but also prevent a perfusion of the bacteria through the intestinal epithelia. Gut microbiota may affect host physiology through metabolic activities and fermentation of nondigestible dietary components and synthesis of vitamins and signaling molecules [7]. Conversely, gut microbiome is influenced by diet and caloric intake [8]. Hence, chronic consumption of a high-fat diet may lead to intestinal barrier defects, enabling microbial metabolites [9] but also gut-derived microbiota [10] to enter into the circulation. Lifestyle-induced weight loss (WL) is regarded as efficient therapy to reverse metabolic syndrome [11]. However, the definition of the microbiome’s role in resolution of metabolic disease following lifestyle-induced WL remains elusive.

Available data about the gut microbiome are mainly derived from genetic approaches focusing on microbiome composition while neglecting actual protein expression and bacterial interaction with the host. The aim of our study was to simultaneously analyze if lifestyle-induced WL for a period of 6 months modulates the gut microbiome and its interaction with the host on a functional level. Hence, we employed a metaproteomics approach to identify proteomic alterations from both the host and the microbiome by using a 24 h protocol [12]. For this protocol, the cells were lysed and the proteins were extracted by phenol extraction in a ball mill. Subsequently, the proteins were digested via FASP digestion and the peptides were analyzed by an LC-MS/MS (Orbitrap Elite™ Hybrid Ion Trap-Orbitrap MS) with a 120 min gradient. The protein identification was carried out with the MetaProteomeAnalyzer software.

To avoid the huge variation between the microbiome of different subjects [13], this study was designed to analyze longitudinal changes in the gut metaproteome in paired feces samples before and after lifestyle-induced WL.

## 2. Materials and Methods

This study was conducted based on a previous study analyzing the impact of lifestyle-induced WL on serum bile acids in individuals with metabolic syndrome [14]. The trial included nonsmoking, nondiabetic men aged between 45 and 55 years with metabolic syndrome as defined by the National Cholesterol Education Program Adult Treatment Panel III guidelines, which is abdominal obesity (waist circumference > 102 cm or BMI > 30 kg/m^2^) combined with at least two of the following criteria: fasting triglyceride (TG) concentration ≥ 1.7 mmol/L; high-density lipoprotein (HDL) cholesterol < 1.05 mmol/L; fasting glucose ≥ 5.6 mmol/L; blood pressure ≥ 130/85 mmHg or treatment for hypertension. Exclusion criteria were smoking, type 2 diabetes mellitus, a history of surgical procedure for WL, severe renal dysfunction (creatinine concentration > 2.0 mg/dL), known liver disease, obesity of known endocrine origin, or inability to walk at least 30 min per day. Participants of the lifestyle-induced WL intervention were advised to lower their calorie intake by 500 kcal/day, to follow a low-carbohydrate diet with preference for carbohydrates with a low glycemic index (as previously described [15]), and to increase their usual daily physical activity by 500 kcal/day but to keep the pulse below 120/min. Moreover, participants recorded body weight daily and received weekly written feedback commenting on their individual weight progress. In total, 59 individuals participated in the lifestyle-induced WL treatment. Thirty-three participants (*n* = 33) provided paired sample sets before and after the WL that were stored at −80 °C and subsequently analyzed by metaproteomics.

Clinical measurements were performed by qualified medical personnel according to standard operating protocols before and after the 6-month intervention period. All blood samples were collected in the morning (8 a.m. to 9 a.m.) from the antecubital vein after a 12 h overnight fast. Glucose was determined in sodium fluoride plasma. Laboratory measurements were performed at the Institute of Clinical Chemistry and Pathobiochemistry, OvGU, Magdeburg, Germany, as described previously [14]. Concentrations of high sensitive CRP (hsCRP) were analyzed using a particle-enhanced immunoturbidimetric assay (Cobas c 501, Roche Diagnostics, Mannheim, Germany).

For the metaproteomic characterization of the microbiomes, we used a previously described and validated workflow [12]. In brief, the cells were lysed and the proteins were extracted by phenol extraction in a ball mill. The protein concentration was quantified by an amido black assay and the proteins were digested via FASP digestion. Subsequently, the peptides were measured by LC-MS/MS (Orbitrap Elite™ Hybrid Ion Trap-Orbitrap; Thermo Fisher Scientific, Bremen, Germany) using a 120 min gradient reversed-phase gradient. For more details, please refer to Appendix A. Protein identification was carried out with the MetaProteomeAnalyzer software, version 3.0 [12] along with the search engines X!Tandem, OMSSA, and Mascot using the following parameters: enzyme trypsin, one missed cleavage, monoisotopic mass, carbamidomethylation (cysteine) as fixed modification, oxidation (methionine) as variable modifications, ±10 ppm precursor and ±0.5 Da MS/MS fragment tolerance, 113C, +2/+3 charged peptide ions, and a false discovery rate adjustment to 1%. The protein database comprised the UniProtKB/SwissProt database (16/01/2019) and the gut metagenome published by Qin et al. (2010) [16]. MS files are accessible in the PRIDE Archive proteome experiment database under the accession number PXD020902. For identified metaproteins lacking a taxonomic or functional annotation, a BLAST search was conducted against UniProtKB/SwissProt [17]. All BLAST hits sharing the best e-value below 10-4 were combined and used to annotate the metaprotein identifications based on the lowest common ancestor approach. Redundant homologous protein identifications were merged to a protein group (hereafter called metaprotein) if they shared at least one peptide identification. Finally, a matrix of all metaproteins over all samples containing the spectral count for each sample and also the annotation information, e.g., NCBI taxonomy, enzyme commission numbers, KEGG orthologies, the UniProtKB reference clusters, and the UniProtKB keywords, was exported and used for data analysis.

This study comprised 33 paired feces samples before and after the 6-month lifestyle-induced WL intervention from individuals with metabolic syndrome. The spectral count for each metaprotein was normalized to the total number of identified spectra for each sample. For statistical analysis, R-Statistics version (1.2.5001) was used. Paired samples were analyzed using the Wilcoxon signed-rank test using the method “wilcox.test”. Power analysis based on 33 samples, significant *p*-values below 0.05, and a realistic standard deviation of 2.66 spectra (for proteins with a spectral count of 5) [12] revealed that the Wilcoxon signed-rank test could detect a twofold change with a power of 96.4%. For further details about reproducibility and robustness of employed method please refer to our previous publication [12]. We defined a cut-off of at least five identified spectra for further results evaluation. For the violin plots the libraries “ggplot2”, “ggstatsplot”, and the method “ggstatsplot:grouped_ggwithinstats” were used. Krona visualization was performed as described in [18].

## 3. Results

### 3.1. Clinical and Laboratory Parameters

In total, 33 subjects of the entire cohort provided fecal samples before and after the 6-month lifestyle-induced WL intervention. Following lifestyle-induced WL participants reduced their individual cardiovascular risk factors such as BMI (−13.7 ± 6.76%), systolic blood pressure (−4.66 ± 9.74%), diastolic blood pressure (−4.36 ± 9.37%), homeostasis model assessment index (HOMA-IR, −46.1 ± 28.6%), total cholesterol (−9.11 ± 14.4%), LDL-cholesterol (−6.85 ± 21.4%), and TG levels (−30.5 ± 34.0%). Moreover, abundance of inflammatory parameters (hsCRP, −39.9 ± 43.0%) as well as liver enzymes (ALAT, −38.3 ± 26.2% and ASAT, −19.6 ± 20.9%) were reduced. Above mentioned changes are given as mean and standard deviation. Details are presented in Figure 1. In general, the changes in these clinical and laboratory measurements are comparable with those obtained in the whole cohort that initially completed the study [14].

### 3.2. Metaproteome Analysis

Metaproteome analysis resulted in an average of 7.234 ± 2.857 spectra (min: 2.904, max: 14.569), 654 ± 177 metaproteins (min: 320, max: 1073), and 53 ± 14 taxonomic families (min: 22, max: 78) for each sample (Appendix A). Taxonomic assignment of all identified spectra (Figure 2, Appendix A) revealed 30.98 ± 11.16% (min: 13.02, max: 63.33) eukaryotic spectra, 22.32 ± 6.15% (min: 9.74, max: 35.65) bacterial spectra, 0.25 ± 0.20% (min: 0.00, max: 1.01) archaeal spectral, and 0.03 ± 0.08% (min: 0.00, max: 0.42) viral spectra across all samples. 32.10 ± 5.32% of all detected spectra (min: 16.82, max: 42.71) could not be assigned to a specific superkingdom (“UNASSIGNED Superkingdom”), whereas for 14.32 ± 3.74% of all spectra (min: 8.76, max: 25.92) (“Unknown MG Entry”) the identified metagenome entries could not be linked to any known protein by BLAST search.

Eukaryotic spectra can be divided into spectra of the host (Hominidae: 3.13 ± 1.30%; min: 1.06, max: 6.75) and spectra of food components (e.g., Fabaceae 0.95 ± 2.31%; min: 0.00, max: 10.70). Bacterial spectra were mainly assigned to the microbial families Bacteroidaceae (3.47 ± 1.88%; min: 0.47, max: 8.82), Clostridiaceae (0.76 ± 0.39%; min: 0.09, max: 1.63), Enterobacteriaceae (0.65 ± 0.53%; min: 0.08, max: 3.55), Bacillaceae (0.59 ± 0.28%; min: 0.18, max: 1.31), and Porphyromonadaceae (0.40 ± 0.33%; min: 0.00, max: 1.33).

Overall, the microbial family richness did not significantly change due to the WL intervention (baseline 39.52 ± 11.91%, after WL 42.85 ± 12.39%) (Appendix A). Regarding the microbial evenness, a significant decrease from 96.29 ± 0.49% (min: 94.99, max: 97.42) to 96.02 ± 0.55% (min: 94.91, max: 97.14) was observed. A significant shift in the ratio between Bacteroidetes to Firmicutes was not observed (before 1.62, after WL 1.61). Regarding individual microbial families, a significant increase was observed for the low abundant families Desulfovibrionaceae (average abundance 0.02%, ratio 3.17, gram negative), Leptospiraceae (average abundance 0.01%, ratio 2.10, gram negative), Syntrophomonadaceae (average abundance 0.004%, ratio 7.50, gram positive), Thermotogaceae (average abundance 0.06, ratio 2.57, gram negative) and Verrucomicrobiaceae (average abundance 0.005%, only after WL, gram negative) (Table 1). Furthermore, an increase in the plant family Rutaceae (average abundance 0.002%, ratio 5.85) and the phage family Siphoviridae (average abundance 0.002%, ratio 4.64) was detected.

A distinction between microbial (bacteria, archaea, fungi, virus) and host metaproteins (metazoa) was made. (Appendix A). The majority of identified host metaproteins corresponded to hydrolysis enzymes (12.0 ± 5.8% of all identified spectra), immunoglobulins (total 2.3 ± 1.7%), structural proteins of the intestinal barrier (2.2 ± 1.1%), neutrophil granulocytes (1.8 ± 0.9%), lysozymes (0.01 ± 0.01%), and angiotensin-converting enzymes (0.01 ± 0.02%) (Figure 3A,B). Lifestyle-induced WL significantly reduced the abundance of ten human metaproteins (Table 2), e.g., pancreatic alpha-amylase (metaprotein 33, ratio 0.39, phylum Chordata), epithelia associated proteins cadherin-1 (metaprotein 370, ratio 0.51, phylum Chordata), HLA class II histocompatibility antigen (metaprotein 1852, ratio only before WL, species Homo sapiens), calcium-activated chloride channel regulator 1 (metaprotein 67, ratio 0.70, class Mammalia), and fibrillin-1 (metaprotein 3996, ratio only before WL, class Mammalia). A decreased abundance was also observed for the neutrophil granulocyte-associated proteins neutrophil gelatinase-associated lipocalin (metaprotein 169, ratio 0.49, species Homo sapiens), α-1-antichymotrypsin (metaprotein 197, ratio 0.63, family Hominidae), and protein S100-A9 (metaprotein 159, ratio 0.60, species Homo sapiens). Furthermore, the abundance of immunoglobulin kappa decreased (metaprotein 212, ratio 0.62, species Homo sapiens; metaprotein 600, ratio 0.38, species Homo sapiens) whereas the abundance of immunoglobulin J chain (metaprotein 1571, ratio 1.75, species Homo sapiens) increased following WL.

Microbial metaprotein spectra were assigned to hydrolysis (0.9 ± 0.6%) as well as to transport proteins for carbohydrates (3.7 ± 1.6%), peptides/amino acids (0.5 ± 0.3%), and glycerol (0.4 ± 0.3%) (Figure 3B,C). Furthermore, transporters for iron, sulfate, and vitamin B12 were detected. However, quantitative assessment of these transporters was imprecise since the different transporters have highly homologous protein sequences. Moreover, microbial metaprotein spectra were assigned to glycolysis (12.5 ± 3.0% of the identified spectra), pentose phosphate pathway (1.7 ± 0.7%), pyruvate degradation (3.2 ± 1.4%), and tricarboxylic acid cycle/succinate fermentation (1.7 ± 0.6%) to acetate (0.2 ± 0.2%), propionate (4.1 ± 1.8%), and butyrate (1.5 ± 0.8%) (Figure 3D). In several samples, evidence of enzymes for the production of ethanol (0.2 ± 0.4%), lactate (0.1 ± 0.1%), format (0.2 ± 0.2%), carbon dioxide (0.1 ± 0.1%), and methane (0.05 ± 0.1%) was found. Concerning amino acid degradation, a multitude of enzymes degrading amino acids to metabolites used in glycolysis or tricarboxylic acid cycle were observed. For example, alanine dehydrogenase (metaprotein 4616, superkingdom Bacteria), which degrades alanine to ammonium and pyruvate. Besides metaproteins involved in metabolization, a multitude of high abundant housekeeping or structural metaproteins, such as 50S ribosomal protein (metaprotein 126, superkingdom Bacteria) or flagellin (metaprotein 47, unknown superkingdom) were detected.

Lifestyle-induced WL was associated with specific alterations of microbial metaproteins that were assigned to hydrolysis and carbohydrate utilization, i.e., increased abundance of endoglucanase A (metaprotein 3789, only in WL, order Clostridiales; metaprotein 8627, ratio only in WL, species Clostridium thermocellum), β-1,4-mannooligosaccharide phosphorylase (metaprotein 1838, ratio only found after WL, species Ruminococcus albus), galactokinase (metaprotein 346, ratio 2.54, unassigned superkingdom), and 5-keto-D-gluconate 5-reductase (metaprotein 1176, ratio 1.58, unassigned superkingdom). Furthermore, lifestyle-induced WL resulted in an increase of microbial flagellin (metaprotein 1745, ratio 7.67, superkingdom Bacteria), rubredoxin (metaprotein 6585, ratio only found after WL, species Clostridium acetobutylicum), and phosphate propanoyltransferase (metaprotein 6265, ratio 1000, species Thermotoga maritima) as well as a decrease of phosphoenolpyruvate carboxykinase (ATP) (metaprotein 87, ratio 0.81, superkingdom Bacteria).

## 4. Discussion

Recent studies suggest that metabolic syndrome is associated with a maladaptive gut microbiome that promotes obesity-associated low-grade inflammation [1] leading to progression of diabetes mellitus and cardiovascular disease [5]. We used a metaproteomic approach to study the effects of lifestyle-induced WL on the gut microbiome and its interaction with the host by simultaneous analyses of host- and microbiota-derived proteins in fecal samples. Since fecal samples can be collected at home by the patient and the metaproteomics workflow can be performed within 24 h [12], the used workflow has a huge clinical potential to monitor changes in the host and its microbiota not only for weight loss but also for other diseases such as inflammatory bowel diseases [2].

Most strikingly, abundance of host metaprotein spectra that are assigned to inflammatory processes, e.g., calprotectin, neutrophil gelatinase-associated lipocalin, as well as α-1-antichymotrypsin (metaproteins 169, 159, 197), were decreased following lifestyle-induced WL. Recent studies indicate that obesity-related microbiome composition has a proinflammatory effect resulting in elevated abundance of inflammatory markers such as calprotectin in fecal samples [19]. Consistent with initially increased values, fecal metaproteomic analysis showed that the proinflammatory changes are reversible upon lifestyle-induced WL. Furthermore, obesity associated gut inflammation is a possible source for increased plasma levels of CRP, calprotectin, and α-1-antichymotrypsin in obese individuals [20,21]. Our results indicate that attenuated gut inflammation may reduce systemic inflammation as suggested by the reduced levels of circulating CRP. In concordance, we observed a reduction of several epithelial host proteins, e.g., cadherin-1 (metaprotein 370), HLA class II histocompatibility antigen (metaprotein 1852), calcium-activated chloride channel regulator 1 (metaprotein 67), and fibrillin-1 (metaprotein 3996), indicating improved epithelia integrity of the gut. Since calprotectin is discussed as a nonspecific marker for colorectal cancer [22], our results suggest that lifestyle-induced WL may also reduce the risk for colon cancer.

Regarding microbiota associated proteins, we detected functional changes. Decreased abundance of metaproteins that correspond to enzymes of the tricarboxylic acid cycle (phosphoenolpyruvate carboxykinase (ATP), metaprotein 87) indicate a reduced fermentation to succinate [23]. This is congruent with attenuated inflammation. Gut microbiota-derived succinate levels are associated with inflammatory processes and increased production of reactive oxygen species that promote local stress, tissue damage, and immune response [24]. Moreover, gut microbiota-derived succinate levels are elevated in insulin-resistant obese individuals [25], indicating maladaptive microbiota–host crosstalk that promotes type 2 diabetes mellitus, cardiovascular disease, and associated inflammation. In addition, we detected an increased abundance of metaproteins that are assigned to microbial enzymes for oxygen detoxification (metaprotein 6585, rubredoxin). Rubredoxin is known to be associated with oxidative stress. A comparative metaproteomic analysis of fecal samples from obese and lean adolescents by Ferrer et al. [4] showed that rubredoxin is only found in fecal samples from lean but not obese individuals. Rubredoxin contains a single [Fe(SCys)_4_] site that is involved in the catalytic reduction of hydrogen peroxide and superoxide [26]. Our results suggest that increased abundance of rubredoxin following lifestyle-induced WL may protect against oxidative stress. Furthermore, we observed an increase of microbial flagellin (metaprotein 1745, ratio 7.67, superkingdom Bacteria). Flagellated bacteria are associated with metabolic syndrome and chronic inflammatory diseases. A host–microbiota interaction implicated in inflammation and obesity is the sensing of flagella through TLR5, which controls motile bacteria by different mechanisms, including the production of antimicrobial peptides and anti-flagella immunoglobulins that regulate the microbiota in the gut [27]. A recent study by Tran et al. [28] analyzed the microbial proteome of feces in response to an obesogenic diet in mice. Most strikingly, levels of fecal flagellin decreased by about five-fold following administration of the obesogenic diet. In line with this, we observed an increased abundance of flagellin following WL. We speculate that due to improved gut epithelial integrity, anti-flagella immunoglobulins decreased after lifestyle-induced WL.

Interestingly, lifestyle-induced WL was associated with reduced abundance of the human pancreatic alpha-amylase (metaprotein 33). Alpha-amylase is secreted for the digestion of starch, promoting resorption of glucose. Of note, insulin has been shown to promote synthesis of pancreatic amylase [29] and secretion of alpha-amylase is enhanced in obese mice [30]. High secretion of pancreatic alpha-amylase is associated with accumulation of visceral fat in animal models when fed a low-fat, high-starch diet [31]. We speculate that the reduced abundance of alpha-amylase following lifestyle-induced WL was caused by reduced insulin levels and by avoidance of carbohydrates with a high glycemic index, which was part of the lifestyle intervention program. This is consistent with improved insulin sensitivity as indicated by reduced HOMA-IR values following lifestyle-induced WL. Next to its digestive function, secretion of alpha-amylase may also play a role in the gut innate immune system and intestinal barrier function. Accordingly, inhibition of exocytosis of the exocrine pancreas is associated with intestinal bacterial outgrowth and dysbiosis [32]. Increased abundance of plant metaproteins (e.g., family Rutaceae) as well as microbial metaproteins that correspond to the hydrolysis of complex carbohydrates (metaproteins 8627, 3789, 1838) and of the enzymes galactokinase (metaprotein 346) and 5-keto-D-gluconate 5-reductase (metaprotein 1176) indicate dietary changes toward higher consumption of fiber-rich foods. Taken together, analysis of fecal host and microbiota proteins revealed that lifestyle-induced WL reduced gut inflammation and induced functional changes in the gut microbiome.

Taxonomic metaproteome analysis revealed that the composition of the core microbiome remained stable following lifestyle-induced WL, confirming comparable studies [33,34]. The ratio between Bacteroidetes to Firmicutes was not affected by lifestyle-induced WL. Our metaproteome analyses indicate that this ratio is not influenced by dietary changes. This conclusion is in accordance with other findings suggesting that the ratio of Bacteroidetes to Firmicutes evolves during different life stages and strongly depends on the individual’s age [35].

Although we did not observe a significant higher microbial family richness, we detected an increase of five lower abundant bacterial families, reflecting adaptations of the microbiome to lifestyle-induced WL. We detected an increased abundance of the family Verrucomicrobiaceae, which includes the mucus degrader Akkermansia muciniphila. Akkermansia muciniphila is reduced in genetically and diet-induced obesity [36]. Moreover, supplementation with Akkermansia muciniphila may improve insulin sensitivity in insulin-resistant obese individuals [37]. We hypothesize that increased abundance of Verrucomicrobiaceae may reflect beneficial crosstalk between the host and gut microbiota following lifestyle-induced WL. The increase of the family Thermotogales may reflect enhanced fermentation of short-chain fatty acids as indicated by increased abundance of phosphotransacetylase (metaprotein 6265). We assume that elevation of Thermotogales was caused by dietary changes and increased consumption of complex carbohydrates with a low glycemic index. Accordingly, the most abundant proteins of this family were the hydrolysis enzymes amylopullulanase (metaprotein 3465) and pullanase (metaprotein 23).

Furthermore, lifestyle-induced WL was associated with an increase in metaproteins assigned to Desulfovibrionaceae, Leptospiraceae, and Syntrophomonadaceae. Of note, these families were not linked to obesity or the gut microbiome before. While Desulfovibrionaceae are increased in mice fed a high-fat diet [38], in other rodents Desulfovibrionaceae increase in response to fasting [39]. The identified metaproteins for Desulfovibrionaceae catalyze dissimilatory sulfate reduction of sulfate to hydrogen sulfide [40], matching to the release of sulfate during mucus degradation. Although, hydrogen sulfide may become toxic, identification of enzymes for the thermodynamically less favorable methanogenesis suggests that its concentration is still quite low. Further studies are required to evaluate the role of Desulfovibrionaceae, Leptospiraceae, and Syntrophomonadaceae on the gut microbial community and on gut inflammation.

The current study had limitations. Study participants were instructed to increase physical activity, to reduce calorie intake, and to perform a low-carbohydrate diet with preference for low glycemic carbohydrates. Beyond these instructions, no special diet, e.g., specific macronutrients, was recommended. Therefore, unfortunately, it is not possible to separate the effects of dietary components on changes in the intestinal metaproteome. Moreover, results from the current study were based on the abundance of identified protein spectra. To evaluate taxonomic composition in detail, a combination with metatranscriptomics and metagenomics would be necessary.

Another shortcoming of the study design was the stringent selection criteria as we only enrolled middle-aged Caucasian males with metabolic syndrome, which precludes generalization of the data to, e.g., females, other racial/ethnic groups, older or younger individuals, or to individuals without metabolic syndrome. Yet at the same time this is a strength as the stringent selection criteria reduced the complexity of the dataset and hence of the analyses. Other advantages of the study were the prospective study design, a strong effect regarding WL as the primary outcome (−13.9%) and robust statistical analyses of paired datasets (before and after intervention). Furthermore, all analyses were conducted in a blinded manner and laboratory measurements were performed according to standard operating protocols, yielding high-quality data.

## 5. Conclusions

In conclusion, lifestyle-induced WL is associated with reduced gut inflammation and functional changes of human and microbial enzymes for carbohydrate hydrolysis. The recently developed metaproteomics workflow has turned out to be a suitable tool to monitor inflammation-associated alterations of host- and microbiota-derived proteins within the gut.

## Figures and Tables

**Figure 1 biomolecules-11-00726-f001:**
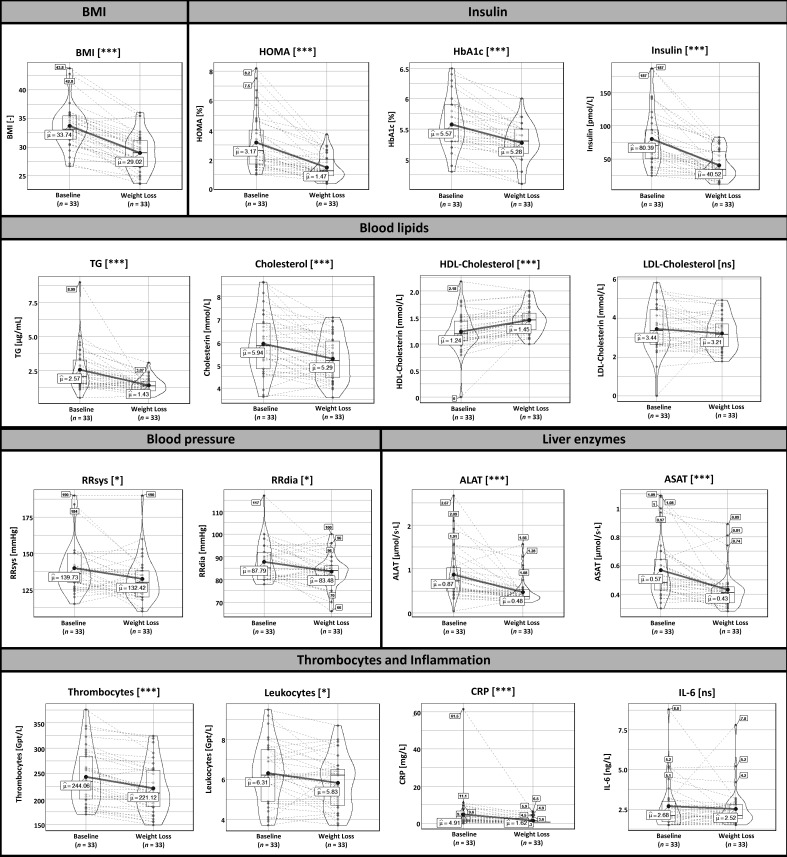
Clinical parameters of individuals with metabolic syndrome at baseline and after the 6-month lifestyle-induced weight loss intervention period. Data are presented as violin plots showing distribution of the data by the inner box with the median (black line) and the interquartile range as well as the kernel probability density of the data by the outer shape. Furthermore, the average (“µ”) is shown and the changes of the values for each patient are indicated by the dashed black lines. The Wilcoxon signed-rank test was used to analyze differences in paired samples (*n* = 33), * *p* < 0.05, *** *p* < 0.001. Abbreviations: ALAT (alanine-aminotransferase), ASAT (aspartate-aminotransferase), BMI (body mass index, CRP (high sensitive C-reactive protein), HDL (high-density lipoprotein cholesterol), HOMA-IR (homeostasis model assessment), LDL (low-density lipoprotein cholesterol), RRDIA (diastolic blood pressure), RRSYS (systolic blood pressure), TG (triglycerides).

**Figure 2 biomolecules-11-00726-f002:**
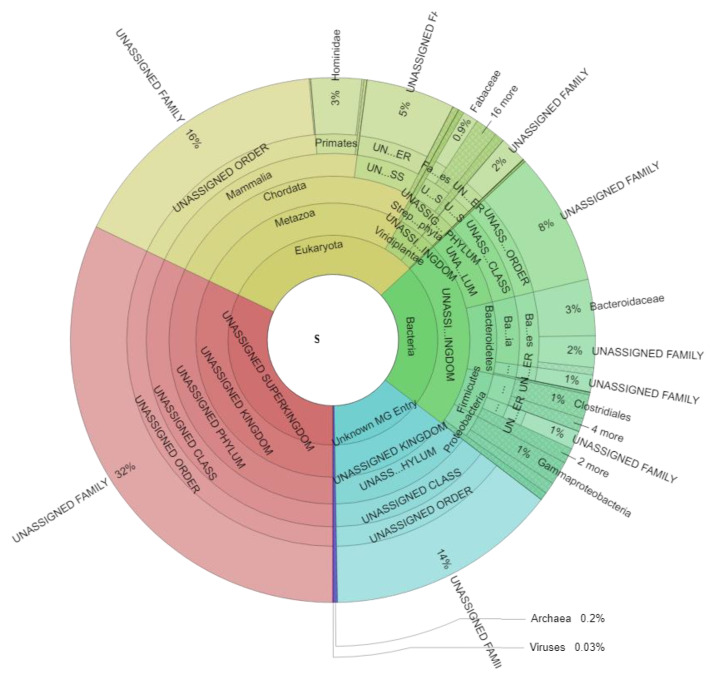
Taxonomic assignment of all identified spectra and significant changes due to weight loss. The Krona plot scheme, 477,469 spectra. Spectra associated with multiple taxonomies and spectra which could not be defined on the specific level are given as “UNASSIGNED” while spectra with no taxonomic assignment are given as “UNKNOWN MG ENTRY”.

**Figure 3 biomolecules-11-00726-f003:**
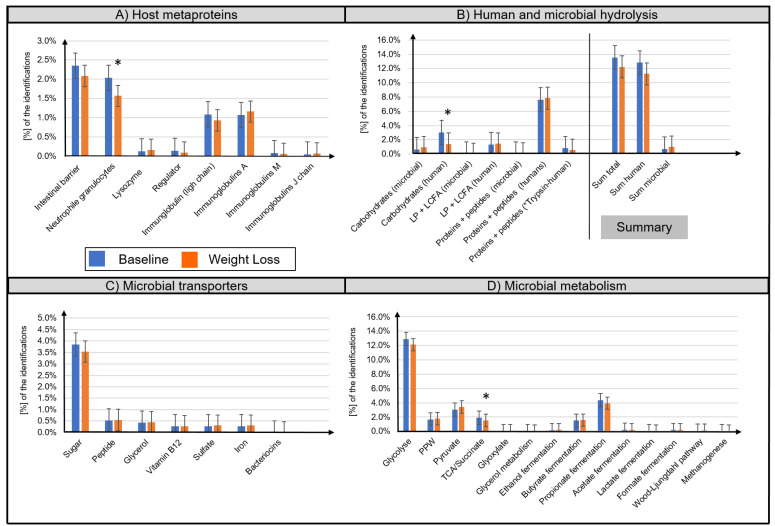
Overview of the functional assignment of identified metaproteins. Identified metaproteins were functionally assigned to microbial and human hydrolysis enzymes (**A**), microbial transporters (**B**), microbial metabolisms (**C**), and host metaproteins (**D**). Data are presented as normalized average spectral abundance. All metaproteins assigned to the kingdom Metazoa were considered as host metaproteins. “Human trypsin” was shown as separated bar in (**B**) since its abundance was potentially increased due to the trypsin used for the tryptic digestion. The Wilcoxon signed-rank test was used to analyze differences in paired samples (*n* = 33, * *p* < 0.05). Bar graphs show mean ± standard deviation.

**Table 1 biomolecules-11-00726-t001:** Summary about significantly altered microbial families.

Family	Abundance Baseline	Abundance after WL	Fold Change	*p*-Value
Thermotogaceae	3.1 × 10^−4^ ± 4.3 × 10^−4^	8.0 × 10^−4^ ± 1.3 × 10^−3^	2.6	0.01
Desulfovibrionaceae	1.1 × 10^−4^ ± 2.1 × 10^−4^	3.4 × 10^−4^ ± 4.8 × 10^−4^	3.2	0.01
Leptospiraceae	7.0 × 10^−5^ ± 2.1 × 10^−4^	1.5 × 10^−4^ ± 2.6 × 10^−4^	2.1	0.04
Syntrophomonadaceae	1.0 × 10^−5^ ± 4.0 × 10^−5^	7.5 × 10^−5^ ± 1.6 × 10^−4^	7.5	0.03
Siphoviridae	7.4 × 10^−6^ ± 3.0 × 10^−5^	3.4 × 10^−5^ ± 6.3 × 10^−5^	4.6	0.03
Rutaceae	4.4 × 10^−6^ ± 2.5 × 10^−5^	2.6 × 10^−5^ ± 5.8 × 10^−5^	5.9	0.05
Verrucomicrobiaceae	Not Detectable	1.0 × 10^−4^ ± 3.7 × 10^−4^	Only in WL	0.04

The Wilcoxon signed-rank test was used to analyze differences in paired samples (*n* = 33). The table shows the average abundance at baseline and after weight loss with the corresponding standard deviation.

**Table 2 biomolecules-11-00726-t002:** Summary of significantly altered metaproteins.

Microbial Metaproteins (ID)	Description	Taxonomy	Average % Abundance	*p*-Value	Fold Change
87	Phosphoenolpyruvate carboxykinase (ATP)	Superkingdom: *Bacteria*	1.5 × 10^−2^	0.03	0.8
1176	5-keto-D-gluconate 5-reductase	Unknown Superkingdom	4.6 × 10^−4^	0.04	1.6
3789	Endoglucanase A	Order: *Clostridiales*	3.3 × 10^−4^	0.00	Only in WL
346	Galactokinase	Unknown Superkingdom	2.3 × 10^−4^	0.00	2.5
8627	Endoglucanase A	Species: *Clostridium thermocellum*	1.3 × 10^−4^	0.04	Only in WL
1745	Flagellin	Superkingdom: *Bacteria*	9.0 × 10^−5^	0.01	7.7
6585	Rubredoxin	Species: *Clostridium acetobutylicum*	4.5 × 10^−5^	0.01	Only in WL
1838	Beta-1,4-mannooligosaccharide phosphorylase	Species: *Ruminococcus albus*	4.5 × 10^−5^	0.04	Only in WL
6265	Phosphate propanoyltransferase	Species: *Thermotoga maritima*	1.4 × 10^−5^	0.04	Only in WL
**Human Metaproteins (ID)**	**Description**	**Taxonomy**	**Average % Abundance**	***p*-Value**	**Fold Change**
33	Pancreatic alpha-amylase	Phylum: *Chordata*	1.9 × 10^−2^	0.00	0.4
67	Calcium-activated chloride channel regulator 1	Class: *Mammalia*	3.8 × 10^−3^	0.01	0.7
159	Protein S100-A9	Species: *Homo sapiens*	2.4 × 10^−4^	0.01	0.6
197	Alpha-1-antichymotrypsin	Family: *Hominidae*	1.7 × 10^−4^	0.02	0.6
212	Immunoglobulin kappa variable 1–33	Species: *Homo sapiens*	8.5 × 10^−4^	0.03	0.6
370	Cadherin-1	Phylum: *Chordata*	6.1 × 10^−4^	0.03	0.5
1571	Immunoglobulin J chain	Species: *Homo sapiens*	6.1 × 10^−4^	0.02	1.7
169	Neutrophil gelatinase-associated lipocalin	Species: *Homo sapiens*	3.3 × 10^−4^	0.05	0.5
600	Immunoglobulin kappa variable 3–15	Species: *Homo sapiens*	2.3 × 10^−4^	0.04	0.4
3996	Fibrillin-1	Class: *Mammalia*	6.3 × 10^−5^	0.02	0.0
1852	HLA class II histocompatibility antigen, DRB1-4 beta chain	Species: *Homo sapiens*	7.6 × 10^−6^	0.04	0.0

Altered human and microbial metaproteins with an abundance > 7.6 × 10^−6^ are summarized. In total, 74 metaproteins were significantly changed (Appendix A). Housekeeping metaproteins or unknown metaproteins were excluded. The taxonomy column shows the lowest confirmed taxonomic rank. The Wilcoxon signed-rank test was used to analyze differences in paired samples (*n* = 33).

## Data Availability

The datasets analyzed during the current study are accessible in the PRIDE Archive proteome experiment database under the accession number PXD020902. The filenames correspond to the ID of the patient (1–33), whereas “C” corresponds to baseline and “ABC” to weight loss.

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
