# Peer review of "Fecal Metaproteomics Reveals Reduced Gut Inflammation and Changed Microbial Metabolism Following Lifestyle-Induced Weight Loss"

_biomolecules, 2021, doi:10.3390/biom11050726_

Round 1
Reviewer 1 Report
The manuscript entitled “Fecal metaproteomics ……Lifestyle induced Weight loss” by Biemann et al explored the alterations in metaproteome of 33 subjects before and after lifestyle-induced weight loss. The study is interesting and is part of another study trial conducted by the authors on serum bile acids associated with lifestyle-induced weight loss. The data will definitely add value to the field of the gut microbiome and its impact on health. However, as eth authors mentioned the study has its own limitations that need to be addressed or considered while referring to this study in the future. The manuscript is written elegantly and easy to understand. I have the following questions and comments.
- I am surprised about the 32 % unassigned superkingdom, what is the reason behind this? Is this the trend observed in other metaproteome studies? Can the unassigned kingdom comprise novel information?
- Can this limitation of the unassigned kingdom be contributed to the experimental or analysis approach?
- Figure 3, change the color for the weight loss group, it is not clear.
- In the previous study published by authors in 2016 authors claimed that the change in bile acids can be contributed to the gut microbiome but I did not find any reference or correlation in this current study to bile acids. Please clarify as it is the continuation of the previous study.
- An increase in Short-chain fatty acids is considered a biomarker for a healthy gut. The authors mentioned an increase in Thermotogales and claimed phosphotransacetylase (metaprotein 625) in this regard, did they find other proteins related to SCFA metabolism?
- My concern over this study design is less control over diet intake. From the authors' precious publication, I understood that the participants were recommended high fiber-rich diets, but the dietary intake was not monitored. Since the gut microbiome varies greatly with the diet I would preferer a controlled trial while studying the gut microbiome and associated omics studies. How do the authors address this, please provide information?
Author Response
We thank the editorial board and the referees for the efforts in reviewing our manuscript (Manuscript ID: biomolecules-1188913). We changed our manuscript according to the suggestions. We now provide more details about the metaproteomics methodology. Furthermore, we improved table 1 and figure 3, and we include a sample description file to the PRIDE submission, so that the samples can be easily tracked and correlated with the manuscript. Moreover, we provide a comprehensive answer about the large percentage of unassigned data. As this discussion is very specific to the metaproteomics community, we decided to exclude it from the manuscript since it requires extensive explanations and detracts from the subject of our study.
- major comment: I am surprised about the 32 % unassigned superkingdom, what is the reason behind this? Is this the trend observed in other metaproteome studies? Can the unassigned kingdom comprise novel information?
The reasons for the 32% of unassigned superkingdoms are homologous peptide identifications associated with different domains, in most cases to Archaea and Bacteria (examples for proteins assigned to Archaea and Bacteria: elongation factor, glyceraldehyde-3-phosphate dehydrogenase, flagellin). Similar observations were also obtained in previous studies [1].
Depending on the grouping strategy of the peptides to proteins and protein groups, the amount of proteins with unassigned superkingdom may differ. We extensively analyzed this issue in the following publications [2,3]. In the present study, we grouped all proteins into a protein group based on overlapping peptide identifications and defined the taxonomy as the common ancestor in the phylogenetic tree. Although this leads to a reduction in the taxonomic classification, it enables strict grouping and better quantification of the protein functions. The latter was favored by us. However, we cannot exclude that the unassigned proteins may comprise beneficial information. A more precise taxonomic assignment of some of these proteins to a deeper taxonomic level, such as phylum or class, would not change the result of our study, which is more related to protein function than to precise taxonomic assignment.
We would like to emphasize that the taxonomically unassigned proteins were still used to compare before and after weight loss.
- major comment: Can this limitation of the unassigned kingdom be contributed to the experimental or analysis approach?
Please see our response to comment 1
- major comment: Figure 3, change the color for the weight loss group, it is not clear.
We followed the reviewer’s suggestion and changed the color for the weight loss group.
- major comment: In the previous study published by authors in 2016 authors claimed that the change in bile acids can be contributed to the gut microbiome but I did not find any reference or correlation in this current study to bile acids. Please clarify as it is the continuation of the previous study.
We thank the reviewer for bringing this point up. In our initial study, we observed changes in serum bile acid composition towards an increased 12α-hydroxylated/non-12α-hydroxylated ratio following weight loss. Possible explanations for observed changes in bile acid pool composition may depend on weight loss associated alterations of hepatic metabolism or on changes of gut microbiota that convey intestinal deconjugation and dihydroxylation of primary bile acids. We identified different metaproteins that are associated with microbial bile acid deconjugation and biotransformation such as formyl-CoA:oxalate CoA-transferase (Metaprotein 8295 and 9542, family: Clostridiaceae), 3α-hydroxy bile acid-CoA-ester 3-dehydrogenase (Metaprotein 1861, family: Clostridiaceae) and NADPH-Fe oxidoreductase subunit beta (Metaprotein 1861, family: unknown). Unfortunately, identified proteins were close to the detection limit of the applied method. Due to the low abundance and since we did not find any changes in these metaproteins after weight loss, we did not report on the bile acid metabolism in the submitted manuscript.
- major comment: An increase in Short-chain fatty acids is considered a biomarker for a healthy gut. The authors mentioned an increase in Thermotogales and claimed phosphotransacetylase (metaprotein 625) in this regard, did they find other proteins related to SCFA metabolism?
We thank the reviewer for that notion. Figure 3D shows the abundance of the proteins related to fatty acid metabolism before and after weight loss. In fact, we did not find any other altered proteins related to SCFA metabolism. However, based on our results we cannot exclude that the SCFA concentration changed after weight loss. Whereas protein abundance correlates with the function, the actual enzyme activity depends also on temperature and substrate concentrations.
- major comment: My concern over this study design is less control over diet intake. From the authors' precious publication, I understood that the participants were recommended high fiber-rich diets, but the dietary intake was not monitored. Since the gut microbiome varies greatly with the diet I would preferer a controlled trial while studying the gut microbiome and associated omics studies. How do the authors address this, please provide information?
We concur with the reviewer that diet is a modifiable factor influencing the composition of the gut microbiota. Participants of the treatment arm were instructed to increase physical activity, reduce calorie intake, and perform a low-carbohydrate diet with preference for low-GI carbohydrates. Beyond these instructions, no special diet, e.g. specific macronutrients, was recommended. Hence, it is unfortunately not possible to link the effects of specific dietary components to the fecal metaproteome.
Our results indicate dietary changes towards higher consumption of fibre-rich foods as it was recommended to the study participants. In this regard, we observed an increased abundance of plant metaproteins (e.g. family Rutaceae) as well as microbial metaproteins that correspond to the hydrolysis of complex carbohydrates (Meta-Protein 8627, 3789, 1838) and the enzymes galactokinase (Metaprotein 346) and 5-keto-D-gluconate 5-reductase (Meta-Protein 1176). This information is given in lines 339-343.
We changed the following text passage to describe the limitation in more detail:
Deleted, lines 383-385: The current findings do not enable us to distinguish which component of the lifestyle modification reduces gut inflammation.
Inserted, lines 383-385: Therefore, unfortunately, it is not possible to separate the effects of dietary components on changes in the intestinal metaproteome.
References
- Heyer, R.; Schallert, K.; Siewert, C.; Kohrs, F.; Greve, J.; Maus, I.; Klang, J.; Klocke, M.; Heiermann, M.; Hoffmann, M.; et al. Metaproteome analysis reveals that syntrophy, competition, and phage-host interaction shape microbial communities in biogas plants. Microbiome 2019, 7, 69, doi:10.1186/s40168-019-0673-y.
- Muth, T.; Behne, A.; Heyer, R.; Kohrs, F.; Benndorf, D.; Hoffmann, M.; Lehtevä, M.; Reichl, U.; Martens, L.; Rapp, E. The MetaProteomeAnalyzer: a powerful open-source software suite for metaproteomics data analysis and interpretation. J. Proteome Res. 2015, 14, 1557–1565, doi:10.1021/pr501246w.
- Heyer, R.; Schallert, K.; Zoun, R.; Becher, B.; Saake, G.; Benndorf, D. Challenges and perspectives of metaproteomic data analysis. J. Biotechnol. 2017, 261, 24–36, doi:10.1016/j.jbiotec.2017.06.1201.

Reviewer 2 Report
The authors presented an interesting study on 33 paired samples before and after the 6-month lifestyle-induced weight-loss intervention from individuals with metabolic syndrome. Metaproteomics was used to compare the taxonomic and functional changes. This paper topic is very interesting, and study is well designed. Below are my comments:
1. Figure 2 is a bit odd to me. In our routine metaproteomic analysis, usually we get more than 95% peptide sequences able to be matched to the superkingdom level and 50% down to the genus level. How would the authors explain the high number of unassigned taxa? I was wondering if the authors try searching their raw data using different metaproteomics database search tools/workflows/databases, such as the MetaLab, would this be improved?
2. How would the authors consider about the increase in microbial rubredoxin and flagellin? These are often found in stress responses.
3. There should be a brief description in the methods about how the samples were processed before MS analysis. It is nice that the authors provided a very detailed workflow as a supplementary note, but key methods should still be briefly summarized in 1-2 sentences in the methods. Similarly, in Introduction, "a 24 h protocol" is too brief and vague to describe what you did. Please expand.
4. Please provide a metafile to the PRIDE archive to clarify what each sample ID stands for.
Minor comments:
1. In abstract, "The recently developed metaproteomics workflow has proven to be a suitable tool for monitoring alterations of host and microbiota proteins in the gut system." This sentence doesn't sound logical here. Maybe replace with a sentence of brief conclusion of significant finding.
2. Page 3 line 3. "Taxonomic and functional characterization of the microbiomes" replace with "metaproteomic characterization of ..."
3. For numbers like "(-13.7 ± 6.76%)", please clarify whether these are Mean+/-SD.
4. Please correct the use of "," and "." in numbers, for example, "1,5E-02" and below in Table 2 should be "1.5E-02". Also, maybe present "E-02" in a better way as well.
5. What was the fragmentation method used for the LC-MS/MS?
6. Table 1 is too big, make it smaller. Also, the column indicating significance is not obvious, add a column name to it will help.
Suggestion for future:
I appreciate the authors giving access to the MS raw data. The samples were well-prepared, as can be seen from the sharp chromatographic peaks. However, for the 150 min samples, the chromatographic peak range was narrow. This seems to me that the gradient was a bit too steep, although the samples were analyzed for 150 min, the effective range was around 1 hr. Consider optimizing the gradient in future before running all samples, this would increase your identification.
Author Response
We thank the editorial board and the referees for the efforts in reviewing our manuscript (Manuscript ID: biomolecules-1188913). We changed our manuscript according to the suggestions. We now provide more details about the metaproteomics methodology. Furthermore, we improved table 1 and figure 3, and we include a sample description file to the PRIDE submission, so that the samples can be easily tracked and correlated with the manuscript. Moreover, we provide a comprehensive answer about the large percentage of unassigned data. As this discussion is very specific to the metaproteomics community, we decided to exclude it from the manuscript since it requires extensive explanations and detracts from the subject of our study.
- major comment: Figure 2 is a bit odd to me. In our routine metaproteomic analysis, usually we get more than 95% peptide sequences able to be matched to the superkingdom level and 50% down to the genus level. How would the authors explain the high number of unassigned taxa? I was wondering if the authors try searching their raw data using different metaproteomics database search tools/workflows/databases, such as the MetaLab, would this be improved?
As it is already pointed out by the reviewer, the database selection and the protein grouping strategy are crucial for taxonomic assignment. Regarding the experimental and bioinformatic workflow, we compared our workflow to the workflow of several other metaproteomics groups and all obtained similar results [1]. We conclude from this study, that the experimental and bioinformatic workflow has little impact on the results. However, we will consider using MetaLab for future studies.
Since we could not create a sample-specific metagenome for all 66 samples, we used a standard reference metagenome catalog for protein identification [2] extended by Uniprot/SwissProt similar to our previous studies [3,4]. We selected this workflow based on a previous study in which we evaluated different databases [4].
Regarding the protein grouping strategy, we decided to group proteins based on overlapping peptide identifications and defined the taxonomy as the common ancestor in the phylogenetic tree. Although this leads to a relatively unspecific taxonomic assignment as observed in other studies [5], it enables a strict grouping and a better quantification of the protein functions. The big issue in the present study is that several shared peptides between Archaeas and Bacterias cause the unprecise taxonomic assignment. Although we agree that Figure 2 may look odd with many taxonomically unassigned proteins, we decided not to hide these data by filtering them out. Regarding the specific taxonomic assignment of the group of the reviewer, we know from collaboration in the biogas field that we get with sample-specific metagenomes assembled to MAGS also 50-60% of unique peptides. We will consider this approach for future studies.
- major comment: How would the authors consider about the increase in microbial rubredoxin and flagellin? These are often found in stress responses.
We thank the reviewer for bringing this point up. We observed an increase of microbial flagellin (Meta-Protein 1745, ratio: 7.67, superkingdom: Bacteria) and rubredoxin (Metaprotein 6585, only found after WL, species: Clostridium acetobutylicum) following lifestyle-induced WL. Rubredoxin is known to be associated with oxidative stress. A comparative metaproteomic analysis of fecal samples from obese and lean adolescents by Ferrer et al. [6] showed that rubredoxin is only found in fecal samples from lean but not obese individuals. Rubredoxin contains a single [Fe (SCys) 4] site that is involved in the catalytic reduction of hydrogen peroxide and superoxide [7]. Our results suggest that increased abundance of rubredoxin following lifestyle-induced WL protects against oxidative stress. Flagellated bacteria are associated with metabolic syndrome and chronic inflammatory diseases. A host-microbiota interaction implicated in inflammation and obesity is the sensing of flagella through TLR5, which controls motile bacteria by different mechanisms, including production of antimicrobial peptides and promoting production of anti-flagella immunoglobulins that regulate the microbiota in the gut [8]. A recent study by Tran et al. [9] analyzed the microbial proteome of feces in response to an obesogenic diet in mice. Most strikingly, levels of fecal flagellin decreased by about 5-fold following administration of the obesogenic diet. In line with this, we observed an increased abundance of flagellin following WL. We speculate that due to improved gut epithelia integrity, anti-flagella immunoglobulins decrease after lifestyle-induced WL.
Inserted, lines 307-325: Rubredoxin is known to be associated with oxidative stress. A comparative metaproteomic analysis of fecal samples from obese and lean adolescents by Ferrer et al. (4) showed that rubredoxin is only found in fecal samples from lean but not obese individuals. Rubredoxin contains a single [Fe (SCys) 4] site that is involved in the catalytic reduction of hydrogen peroxide and superoxide (26). Our results suggest that increased abundance of rubredoxin following lifestyle-induced WL may protect against oxidative stress. Furthermore, we observed an increase of microbial flagellin (Meta-Protein 1745, ratio: 7.67, superkingdom: Bacteria). Flagellated bacteria are associated with metabolic syndrome and chronic inflammatory diseases. A host-microbiota interaction implicated in inflammation and obesity is the sensing of flagella through TLR5, which controls motile bacteria by different mechanisms, including the production of antimicrobial peptides and anti-flagella immunoglobulins that regulate the microbiota in the gut (27). A recent study by Tran et al. (28) analyzed the microbial proteome of feces in response to an obesogenic diet in mice. Most strikingly, levels of fecal flagellin decreased by about 5-fold following administration of the obesogenic diet. In line with this, we observed an increased abundance of flagellin following WL. We speculate that due to improved gut epithelial integrity, anti-flagella immunoglobulins decrease after lifestyle-induced WL.
- major comment: There should be a brief description in the methods about how the samples were processed before MS analysis. It is nice that the authors provided a very detailed workflow as a supplementary note, but key methods should still be briefly summarized in 1-2 sentences in the methods. Similarly, in Introduction, "a 24 h protocol" is too brief and vague to describe what you did. Please expand.
We agree with the reviewer that the method should be described in more detail. We added the following information in the introduction and in the material and methods section to describe the workflow.
Inserted, lines 72-76: For this protocol, the cells were lysed and the proteins were extracted by phenol extraction in a ball mill. Subsequently, the proteins were digested by FASP digest, and the peptides were analyzed by an LC-MS/MS (Orbitrap Elite™ Hybrid Ion Trap-Orbitrap MS) with a 120 min gradient. The protein identification was carried out with the Meta-Proteome Analyzer [4].
Inserted, lines 108-113: For the metaproteomic characterization of the microbiomes, we used a previously described and validated workflow [4]. In brief, the cells were lysed, and the proteins were extracted by phenol extraction in a ball mill. The protein concentration was quantified by amido black assay, and the proteins were digested by FASP digest. Subsequently, the peptides were measured by LC-MS/MS (Orbitrap Elite™ Hybrid Ion Trap-Orbitrap) using a 120 min gradient reversed-phase gradient.
- major comment: Please provide a metafile to the PRIDE archive to clarify what each sample ID stands for.
We added in the Pride description under “Data Availability Statement”.
Inserted, lines 425-427: The filenames correspond to the ID of the patient (1-33), whereas “C” corresponds to baseline and “ABC” to weight loss.
- minor comment: In abstract, "The recently developed metaproteomics workflow has proven to be a suitable tool for monitoring alterations of host and microbiota proteins in the gut system." This sentence doesn't sound logical here. Maybe replace with a sentence of brief conclusion of significant finding.
We agree with the reviewer and we replaced the sentence with a brief conclusion.
Changed, lines 39-40: The Metaproteomics workflow has proven to be a suitable method for monitoring inflammatory changes in the fecal metaproteome.
- minor comment: Page 3 line 3. "Taxonomic and functional characterization of the microbiomes" replace with "metaproteomic characterization of ..."
We replaced the wording according to the reviewer´s suggestion.
Changed, line 108-109: For the metaproteomic characterization of the microbiomes, we used a previously described and validated workflow [12].
- minor comment: For numbers like "(-13.7 ± 6.76%)", please clarify whether these are Mean+/-SD.
We thank the reviewer for that notion and included this information now.
Inserted, lines 154-155: Above mentioned changes are given as mean and standard deviation. Details are presented in figure 1.
- minor comment: Please correct the use of "," and "." in numbers, for example, "1,5E-02" and below in Table 2 should be "1.5E-02". Also, maybe present "E-02" in a better way as well.
We thank the reviewer for that notion and correct the use of "," and "." in numbers in Table 2. Furthermore, we changed “E-02” into 10-2. The same changes were made in Table 1.
- minor comment: What was the fragmentation method used for the LC-MS/MS?
We used collision-induced dissociation. This information can be found in the supplementary file with a detailed description of the entire workflow.
- minor comment: Table 1 is too big, make it smaller. Also, the column indicating significance is not obvious, add a column name to it will help.
We agree with the reviewer that table 1 is too large. We have now changed table 1 according to table 2.
Suggestion for future:
I appreciate the authors giving access to the MS raw data. The samples were well-prepared, as can be seen from the sharp chromatographic peaks. However, for the 150 min samples, the chromatographic peak range was narrow. This seems to me that the gradient was a bit too steep, although the samples were analyzed for 150 min, the effective range was around 1 hr. Consider optimizing the gradient in future before running all samples, this would increase your identification.
We thank the reviewer for this helpful advice. We are impressed that the reviewer took the time to check the MS measurements. We will have a meeting next Monday with our team to decide how we can optimize the gradient.
References
- van den Bossche, T.; Kunath, B.J.; Schallert, K.; Schäpe, S.S.; Abraham, P.E.; Armengaud, J.; Arntzen, M.Ø.; Bassignani, A.; Benndorf, D.; Fuchs, S.; et al. Critical Assessment of Metaproteome Investigation (CAMPI): A Multi-Lab Comparison of Established Workflows, 2021.
- Qin, J.; Li, R.; Raes, J.; Arumugam, M.; Burgdorf, K.S.; Manichanh, C.; Nielsen, T.; Pons, N.; Levenez, F.; Yamada, T.; et al. A human gut microbial gene catalogue established by metagenomic sequencing. Nature 2010, 464, 59–65, doi:10.1038/nature08821.
- Lehmann, T.; Schallert, K.; Vilchez-Vargas, R.; Benndorf, D.; Püttker, S.; Sydor, S.; Schulz, C.; Bechmann, L.; Canbay, A.; Heidrich, B.; et al. Metaproteomics of fecal samples of Crohn's disease and Ulcerative Colitis. J. Proteomics 2019, 201, 93–103, doi:10.1016/j.jprot.2019.04.009.
- Heyer, R.; Schallert, K.; Büdel, A.; Zoun, R.; Dorl, S.; Behne, A.; Kohrs, F.; Püttker, S.; Siewert, C.; Muth, T.; et al. A Robust and Universal Metaproteomics Workflow for Research Studies and Routine Diagnostics Within 24 h Using Phenol Extraction, FASP Digest, and the MetaProteomeAnalyzer. Front. Microbiol. 2019, 10, 1883, doi:10.3389/fmicb.2019.01883.
- Heyer, R.; Schallert, K.; Siewert, C.; Kohrs, F.; Greve, J.; Maus, I.; Klang, J.; Klocke, M.; Heiermann, M.; Hoffmann, M.; et al. Metaproteome analysis reveals that syntrophy, competition, and phage-host interaction shape microbial communities in biogas plants. Microbiome 2019, 7, 69, doi:10.1186/s40168-019-0673-y.
- Ferrer, M.; Ruiz, A.; Lanza, F.; Haange, S.-B.; Oberbach, A.; Till, H.; Bargiela, R.; Campoy, C.; Segura, M.T.; Richter, M.; et al. Microbiota from the distal guts of lean and obese adolescents exhibit partial functional redundancy besides clear differences in community structure. Environ. Microbiol. 2013, 15, 211–226, doi:10.1111/j.1462-2920.2012.02845.x.
- Lumppio, H.L.; Shenvi, N.V.; Summers, A.O.; Voordouw, G.; Kurtz, D.M. Rubrerythrin and rubredoxin oxidoreductase in Desulfovibrio vulgaris: a novel oxidative stress protection system. J. Bacteriol. 2001, 183, 101–108, doi:10.1128/JB.183.1.101-108.2001.
- Cullender, T.C.; Chassaing, B.; Janzon, A.; Kumar, K.; Muller, C.E.; Werner, J.J.; Angenent, L.T.; Bell, M.E.; Hay, A.G.; Peterson, D.A.; et al. Innate and adaptive immunity interact to quench microbiome flagellar motility in the gut. Cell Host Microbe 2013, 14, 571–581, doi:10.1016/j.chom.2013.10.009.
- Tran, H.Q.; Mills, R.H.; Peters, N.V.; Holder, M.K.; Vries, G.J. de; Knight, R.; Chassaing, B.; Gonzalez, D.J.; Gewirtz, A.T. Associations of the Fecal Microbial Proteome Composition and Proneness to Diet-induced Obesity. Mol. Cell. Proteomics 2019, 18, 1864–1879, doi:10.1074/mcp.RA119.001623.
Round 2
Reviewer 2 Report
The authors have addressed all my concerns.